# Thrombin Is an Effective and Safe Therapy in the Management of Bleeding Gastric Varices. A Real-World Experience

**DOI:** 10.3390/jcm10040785

**Published:** 2021-02-16

**Authors:** Sarah-Louise Gillespie, Norma C. McAvoy, Diana E. Yung, Alexander Robertson, John N. Plevris, Peter C. Hayes

**Affiliations:** 1Centre for Liver and Digestive Disorders, Royal Infirmary of Edinburgh, Edinburgh H16 4SA, UK; norma.mcavoy@ed.ac.uk (N.C.M.); Diana.Yung@nhslothian.scot.nhs.uk (D.E.Y.); Alexander.Robertson@nhslothian.scot.nhs.uk (A.R.); j.plevris@ed.ac.uk (J.N.P.); p.hayes@ed.ac.uk (P.C.H.); 2College of Medicine and Veterinary Medicine, University of Edinburgh, Edinburgh EH8 9YL, UK

**Keywords:** thrombin, varices, portal hypertension, cirrhosis

## Abstract

Variceal haemorrhage is a severe complication of liver disease with high mortality. Human recombinant thrombin has gained popularity in the management of variceal haemorrhage. We report on the use of thrombin for gastric and ectopic varices at a regional tertiary care centre. This was a retrospective observational study. Patients with portal hypertension who received endoscopic injection of recombinant thrombin were identified and data collected on haemostasis and rebleeding rates, complications and mortality. Patients were grouped by indication for thrombin injection: gastric/oesophageal/ectopic varices and endoscopic band ligation (EBL)-induced ulceration. 155 patients (96M/59F, mean age 58.3 years) received endoscopic thrombin injection. Mean volume of thrombin injected at index endoscopy was 9.5 ml/2375IU. Initial haemostasis was achieved in 144 patients (92.9%). Rebleeding occurred in a total of 53 patients (36.8%) divided as follows: early rebleeding (<5 days from index endoscopy)—26 patients (18%); rebleeding within 30 days—42 patients (29.1%); delayed rebleeding (> 30 days)—11 patients (7.6%). There was statistically significant difference in rate of initial haemostasis between Child-Pugh A/B patients vs Child-Pugh C (*p* = 0.046). There was no significant difference in rebleeding rates between different indication groups (*p* = 0.78), by presence of cirrhosis or by Child-Pugh Score. All-cause mortality at 6 weeks was 18.7%; 1-year mortality 37.4% (median follow-up 48 months). There was no significant difference in mortality between groups (*p* = 0.37). No significant adverse events or complications were reported. Thrombin is effective and safe for gastric varices and other portal-hypertension-related bleeding including oesophageal varices, ulcers secondary to EBL and ectopic varices.

## 1. Introduction

Variceal haemorrhage is a severe complication of liver disease with high morbidity and mortality. Overall, the prevalence of gastro-oesophageal varices is between 40–85% in patients with cirrhosis, which is proportional to the severity of the underlying liver disease. Gastric varices are less common and their prevalence in patients with portal hypertension is estimated at around 20%, with bleeding from gastric varices thought to account for 10–30% of all variceal bleeding [1,2]. Although bleeding from gastric varices is less common compared with oesophageal varices, this tends to be more severe with higher mortality [1,3,4].

Compared with the extensive research into the management of bleeding oesophageal varices, evidence for the best way to treat gastric and ectopic varices is much more limited. Current British Society of Gastroenterology (BSG) guidelines recommend intra-variceal injection of either N-butyl-cyanoacrylate (glue) or thrombin for the endoscopic management of acute gastric variceal haemorrhage. Large studies regarding the effectiveness and safety of thrombin are lacking, and injection therapy with glue is favoured by many centres, resulting in less familiarity with thrombin. Whilst cyanoacrylate has been demonstrated to be effective in achieving haemostasis and preventing re-bleeding, it does have potential for serious thrombotic complications including stroke and pulmonary embolism [5,6,7]. The other main approach to acutely bleeding gastric varices is interventional radiology (IR), primarily transjugular portosystemic stent-shunt (TIPSS); this is generally reserved for bleeding refractory to endoscopic therapy or for secondary prevention of rebleeding [8]. As not all centres have 24-h access to interventional radiology, and these techniques are not technically possible in all patients, endoscopic therapeutic options remain vitally important.

The use of bovine and subsequently recombinant human thrombin has been reported in the management of oesophageal, gastric and duodenal varices. Our group has previously reported on the use of human thrombin in managing bleeding gastric and ectopic varices, showing that it is effective in achieving haemostasis with a low re-bleeding rate of 10.8% [9]. Similar efficacy has been reported by other groups including Smith et al. [10], with some variation in the observed rebleeding rates [10,11,12]. Major advantages of thrombin therapy include excellent safety in particular regarding embolic complications and easiness of use.

The aim of this study is to report our real-world experience of the use of human thrombin in the management of portal hypertension-related bleeding, in particular gastric and ectopic varices, in a large cohort of patients in a tertiary referral centre for liver disease.

## 2. Materials and Methods

This was a retrospective observational study. Patients who underwent endoscopy between 1 May 2011 and 11 January 2019 inclusive were identified from the local endoscopy database. Patients were included who had received injection of recombinant thrombin during the index endoscopy. Individual patient data were manually collected from electronic patient records. Exclusion criteria were: age <18 years and therapy for a non-variceal indication.

### 2.1. Patient Groups

Patients were grouped by indication for thrombin injection: 1. Gastric varices; 2. Oesophageal varices; 3. Endoscopic band ligation (EBL)—induced ulceration; 4. Ectopic varices. Ectopic varices were classified as varices in locations other than the gastro-oesophageal region, in this study comprising patients with duodenal, stomal or rectal varices. The presence or absence of cirrhosis was determined from the patients’ electronic clinical records including imaging and blood results. Child-Pugh score [13] and MELD score [14] for cirrhotic patients were recorded using values from date of hospital admission.

Gastric varices were subclassified based on endoscopic appearance by Sarin’s classification [1,2] into gastro-oesophageal varices (GOV1—extending along the lesser curvature; GOV2—extending into the fundus) and isolated gastric varices (IGV1—fundal; IGV2—gastric body and antrum). Cross sectional imaging was not performed in the majority of patients to further characterise intra-abdominal vasculature.

Data regarding the number of patients with known varices or previous endoscopic therapy prior to the index admission was not fully available. Transfusion data was also not fully available for inclusion. Accordingly, incomplete data were not included in this report.

### 2.2. Endoscopic Therapy

All patients had an upper gastrointestinal endoscopy (OGD) or flexible sigmoidoscopy performed by a trained endoscopist following local protocol after any initial resuscitation.

Variceal haemorrhage was defined as active bleeding or stigmata of recent bleeding, e.g., adherent clot or red spots, from oesophageal, gastric or ectopic varices. EBL-induced ulceration was defined as evidence of active bleeding (spurting or oozing) from an oesophageal ulcer formed following slippage of a recently placed band.

Thrombin was utilised as the primary haemostatic agent according to our standard protocol in preference to N-butyl-cyanoacrylate. Our group has previously demonstrated that rebleeding is rare after 3 sessions of thrombin injection even without visual eradication of gastric varices at weekly or two weekly intervals [9].

When endoscopic therapy failed, balloon tamponade was utilised as a temporising measure and further endoscopic therapy or a transjugular intrahepatic portosystemic shunt (TIPSS) procedure was performed at the discretion of the clinical team. If indicated, repeat endoscopy was arranged for further thrombin injection at 1–2 weekly intervals until endoscopist decision that no further therapy was required. This decision was based on the varices appearing well covered by mucosa with no stigmata of recent haemorrhage. The number of endoscopy sessions, total volume of thrombin injected, and visual eradication of varices were documented.

### 2.3. Protocol for Thrombin Therapy

Recombinant human thrombin was sourced from off-label use of the thrombin component from Tisseel (Baxter Inc, Glenview, IL, USA) and subsequently Floseal haemostatic matrix (Baxter Inc, USA). The per unit price for this kit (as of 2021) was £287.83. Each 2500IU vial of thrombin powder was reconstituted with 10mL sodium chloride 0.9% by gently swirling until completely dissolved, to give a final concentration of 250IU/mL. Thrombin was injected using a standard 23GA sclerotherapy needle directly into the varix or bleeding banding ulcer. The needle was held in place for up to 10 s before retracting to reduce the risk of bleeding from the puncture site. This was performed using multiple injections of 1–5 mL thrombin per injection site, with a mean of 10 mL in a single session.

### 2.4. Statistical Analysis

Survival and rebleeding data for the 4 patient groups were plotted as Kaplan Meier survival curves and Log Rank tests performed for comparison. Outcomes by Child-Pugh score and presence/absence of cirrhosis were compared by Chi-Square test and rebleeding rates plotted as Kaplan Meier curves. Median follow up was calculated as median time to censoring, regardless of survival status. Statistical analysis was performed using SPSS (IBM SPSS Statistics for Windows, IBM Corp, Version 25.0. Armonk, NY, USA). A *p*-value of <0.05 was taken to denote statistical significance.

### 2.5. Ethics

This study was conducted in accordance with UK research ethics guidelines. The local ethics committee considered that formal ethical approval was not necessary, as the study was considered to be a retrospective audit.

## 3. Results

### 3.1. Demographics

155 patients received endoscopic therapy with recombinant thrombin during the study period. 96 patients (62%) were male with a median age of 58 years at presentation (range: 27–84 years). The underlying cause of portal hypertension was: cirrhosis in 138 patients (89%); non-cirrhotic secondary to portal vein or splenic vein thromboses in 15 patients (10%) and nodular regenerative hyperplasia in one patient. One patient had gastric varices secondary to extensive gastric adenocarcinoma and no definite evidence of portal hypertension.

In the cirrhotic cohort, the underlying aetiology was predominantly alcohol-related liver disease or non-alcoholic fatty liver disease. Among cirrhotic patients, 115/138 (83.3%) were Child-Pugh grade B or C; mean and median MELD were 18.7 and 16 respectively. Two patients were post orthotopic liver transplant and data were not available for a further six patients. Full details of patient cohorts are given in Table 1. Platelet count at admission for the index bleed for all groups is given in Appendix A.

### 3.2. Indications for Thrombin Therapy

Indications for endoscopic therapy were: emergency OGD due to acute upper GI bleeding (136 patients, 88%), or lower GI bleeding due to rectal varices (eight patients, 5%); elective OGD, with evidence of bleeding/high risk varices at OGD or planned thrombin injection to previously detected gastric varices (11 patients, 7%).

112 patients (72.2%) were treated for gastric varices. Other indications included ulceration secondary to prior endoscopic band ligation (EBL), oesophageal varices and ectopic varices (duodenal, stomal and rectal) (Table 2). Those with gastric varices were categorised at the time of endoscopy as per Sarin’s classification. 91 patients (81%) had either GOV1 or GOV2 compared with 21 who had isolated gastric varices (IGV1 and IGV2). EBL of oesophageal varices was performed at index endoscopy in 64 patients (39.8%).

The mean number of endoscopic procedures per patient was 1.9 (range: 1–9, SD 1.3), median 1.0. A mean volume of 9.5ml (range: 2–26mL), equating to 2375IU of thrombin was injected at the index endoscopy with a mean volume of 18.3ml (range: 2–101mL) injected in total over all procedures. Endoscopic ultrasound (EUS), including mini-probe (mEUS), was performed to confirm eradication or to facilitate targeted therapy in 11 cases.

### 3.3. Efficacy of Thrombin Therapy

Across all groups, initial haemostasis utilising thrombin was achieved in 144 (92.9%) patients. Immediate haemostasis was not achieved in 11 patients; balloon tamponade with a Sengstaken-Blakemore tube (SBT) was used in all cases as a temporising measure. SBT removal at 24–48 h was followed by rescue therapy with further thrombin injection (1 patient, 9%), OV banding (6 patients, 55%) and TIPSS insertion/upsizing (4 patients, 36%).

Rebleeding episodes were identified as a further episode of overt GI blood loss (haematemesis or melaena), or a drop in haemoglobin with no other cause identified and with clinical suspicion of further bleeding. Rebleeding was divided into early (<5 days from index endoscopy), 30-day rebleeding rate (bleeding 0–30 days post endoscopy) and delayed rebleeding (>30 days) to allow for comparison.

Rebleeding occurred in a total of 53 patients (36.8%) as follows: early rebleeding–26 patients (18%); rebleeding within 30 days–42 patients (29.1%); delayed rebleeding (range 39–696 days)–11 patients (7.6%). In the early rebleeding group, 24 (92%) had a repeat endoscopy to confirm that this was a true rebleeding event rather than prolonged melaena from the index bleed. Rescue TIPSS insertion was performed in 15 patients (9.7%) following index bleed or rebleeding episode, with a further 5 patients (3.2%) undergoing upsizing of an existing TIPSS. There was no significant difference in rebleeding rates demonstrated between different indications for thrombin injection (Log Rank test, *p* = 0.85). Figure 1 shows rebleeding rates across the four patient groups.

### 3.4. Mortality

All-cause mortality at six weeks (as per Baveno VI consensus [15] was 18.7% with one year mortality 37.4% (Figure 2a). Median follow up was 48 months (range 1–96; IQR 41). There was no significant difference in mortality rates between groups (Log Rank test, *p* = 0.46) (Figure 2b). 28 patients (31%) died directly as a result of the index bleeding episode or precipitated by this during the same hospital admission. Overall, GI haemorrhage was cited as a cause of death in 33% of cases (29 patients).

### 3.5. Adverse Events

There were no reported clinically significant adverse events or complications resulting from thrombin therapy for any indication over the duration of follow up in our patient group. A transient drop in blood pressure was occasionally seen immediately after thrombin injection which resolved in all cases without any specific treatment.

### 3.6. Subgroup Analysis

#### 3.6.1. Gastric Varices

In the gastric varices group, initial haemostasis using thrombin was achieved in 105 (93.8%) patients. Overall rebleeding rate was 37.1%: early rebleeding–18 patients (17.1%); 30-day rebleeding rate–29 patients (26.8%); delayed rebleeding (range 39–696 days)-10 patients (9.5%). Of those patients who rebled, 22 (56.4%) were successfully managed with further endoscopic therapy with thrombin. 11 (28%) were managed with rescue TIPSS insertion or upsizing. Three patients (7.7%) underwent splenic artery embolization. Reasons for attempting further endoscopic therapy rather than TIPSS in patients who rebled included the presence of splenic or portal vein thromboses making the procedure technically challenging, and the presence of hepatic encephalopathy.

One patient with gastric varices secondary to cirrhosis and splenic vein thrombosis had three early rebleeding episodes all managed with thrombin injection due to being unsuitable for TIPSS. On the fourth rebleeding episode they were successfully managed with intra-variceal injection of N-butyl-cyanoacrylate with no further bleeding following this.

The mean number of endoscopies per patient was 2.1 (Range: 1–9, SD 1. Median 2). Gastric varices were deemed visually eradicated following therapy in only 18 patients (16.1%). 6-week mortality was 19.6%, with 1-year mortality 34.8%.

#### 3.6.2. Oesophageal Varices and EBL-Induced Ulceration

In the EBL-induced ulceration group, initial haemostasis was achieved in 17 patients (89%). The overall rebleeding rate was 29.4%: early rebleeding–2 patients (11.8%); 30-day rebleeding rate–5 patients (29.4%); no delayed rebleeding. In this group, six-week mortality was 11.1%, with a one-year mortality rate of 42.1%.

In the oesophageal varices group, initial haemostasis was achieved in 12 patients (85.7%). The overall rebleeding rate was 33.3%: early rebleeding–3 patients (25%); 30-day rebleeding rate–4 patients (33.3%); no delayed rebleeding. Standard therapy in this group would usually be with EBL. Six patients (42.9%) received EBL in addition to thrombin at the index endoscopy with a further two patients having had EBL at a prior endoscopy within the preceding seven days. In cases where EBL was not utilised this was either due to the finding of a suspected Mallory-Weiss Tear overlying an oesophageal varix, or technical difficulty banding due to the OV location. Mortality rate at six weeks was 28.6%, with one-year mortality of 57.1%.

#### 3.6.3. Ectopic Varices

The distribution of ectopic varices in this group included rectal, stomal and duodenal, shown in Table 2. In the ectopic varices group, initial haemostasis was achieved in all 10 patients (100%). The overall rebleeding rate was 50%: early rebleeding–3 patients (33.3%); 30-day rebleeding rate 4 patients (40%); delayed rebleeding (at 157 days)–1 patient (10%). In this group, no patients died within six weeks of the index bleed, with a one-year mortality of 10%.

#### 3.6.4. Results by Presence or Absence of Cirrhosis

In our cohort, 138 patients were known to be cirrhotic whilst 17 were not, as per Table 1. Table 3 compares outcomes between cirrhotic and non-cirrhotic patients. Chi-square tests were largely unavailable for these outcomes due to the small sample size of the non-cirrhotic group. There was no significant difference in rebleeding rate between patients with cirrhosis and those with non-cirrhotic portal hypertension.

#### 3.6.5. Results by Child-Pugh Score

Table 4 gives a breakdown of outcomes for cirrhotic patients stratified by Child-Pugh score. Data were available for 130 patients. The causes of index bleed were not significantly different between the three groups however there was a statistically significant difference in whether initial haemostasis was achieved between patients who were Child-Pugh A, B and C (*p* = 0.046). There was no significant difference between the percentages of patients in each group who re-bled following initial haemostasis and no significant difference in timing or cause of re-bleed. Figure 1b shows rebleeding stratified by Child-Pugh score. The proportion of patients alive or dead at time of follow-up was significantly different between the groups (*p* = 0.021).

## 4. Discussion

Bleeding from gastric varices accounts for 10–30% of all variceal bleeding however, compared to oesophageal varices, there is limited evidence on optimal management. Endoscopic therapy and interventional radiology techniques, namely TIPSS or balloon-occluded retrograde transvenous obliteration (BRTO), are recommended therapeutic options for acute gastric variceal haemorrhage and prevention of rebleeding [16,17].

Current endoscopic options for gastric varices include endoscopic band ligation (EBL) and direct injection of tissue adhesives or thrombin. EBL is well-established for the treatment of bleeding oesophageal varices and can be utilised for GOV1 gastric varices as these are considered an extension of OVs [17]. The evidence for its use for GOV2 and isolated gastric varices remains far more unclear however.

Tissue adhesives (glue) including N-butyl-cyanoacrylate are currently recommended as first line endoscopic therapy for acute gastric variceal bleeding [15,17]. A number of non-randomised studies have demonstrated initial haemostasis rates of over 90% with rebleeding rates of 22–37% [5,6,18,19,20,21,22,23]. Two randomised controlled trials by Lo [5] and Tan et al. [19] have compared N-butyl-cyanoacrylate with EBL with both showing lower rebleeding rates in the cyanoacrylate groups; however there was no demonstrated difference in mortality [5,19,24]. Complications and adverse events secondary to injection of tissue adhesives have been documented in a number of case reports and small case series. These are predominantly embolic in nature with reports including the development of pulmonary emboli, splenic infarction and cerebral stroke following glue administration, however localised and technical complications can also occur [25,26,27]. The technique when administering glue is critical to avoid irreparable damage to the endoscope or accidental adhesion of the injection needle to the varix. This has the potential to cause massive bleeding and fatalities have been reported [28].

Thrombin acts as a haemostatic agent by converting fibrinogen to a fibrin clot and was first described as a management option for gastric varices in 1947 [29]. Originally bovine thrombin was used, but due to the concerns regarding potential prion disease transmission, recombinant human thrombin is now utilised. A number of uncontrolled observational studies have demonstrated the use of thrombin as an effective alternative to glue injection, although the data remains limited with small numbers of patients [9,11,12,30,31,32]. Studies using bovine thrombin by Williams [12] and Przemioslo et al. [11] demonstrated haemostasis rates of 94–100% with rebleeding rates of 27% and 18% respectively, but were limited by relatively short follow-up periods. Ramesh et al. [31], also utilising bovine thrombin, demonstrated a haemostasis rate of 92% with no rebleeding reported over the follow-up period of 25 months in a group of 13 patients.

Our group has previously published a case series of 37 patients treated with human thrombin for gastric and ectopic varices at our regional tertiary care centre, demonstrating an initial haemostasis rate of 100% and rebleeding rate of 10.8% [9]. The mortality rate in this group was also extremely low at only 2.7% over a median follow-up period of 22 months. More recently, Smith et al. [10] reported outcomes from their group of 30 patients receiving human thrombin for gastric varices, with 20 receiving it for active bleeding. They demonstrated initial haemostasis in 90% of patients with a rebleeding rate of 55%, which included those in whom haemostasis was not achieved. In this cohort, 20% required rescue TIPSS for failure to control initial bleeding or re-bleeding, and thrombin was therefore suggested as a bridge to more definitive therapy. Another study by Jhajharia et al. [33] of 20 patients demonstrated initial haemostasis in 100% following thrombin injection, with a rebleeding rate of only 5% over the duration of follow up and no mortality over a mean follow up of 16.8 months.

Lo et al. [34] recently reported on the first prospective randomised trial of thrombin versus cyanoacrylate in acute gastric variceal haemorrhage. 33 patients were randomised to the thrombin arm with 35 randomised to glue injection; the primary outcome was the development of injection-induced gastric ulceration. Initial haemostasis rates of 90% were achieved in the thrombin group and 90.9% in the cyanoacrylate group. There was no difference in treatment failure rates at five days (6.1% in the thrombin arm vs 5.7% in the glue arm, *p* > 0.99) and mortality rates at six weeks also showed no significant difference (3.0% and 2.9% respectively). The complication rate was significantly higher in the glue group (51.4%) compared with the thrombin group (12.1%).

Our initial haemostasis rate of 93.8% for management of gastric varices is comparable with other retrospective observational studies, demonstrating the effectiveness of thrombin in acute bleeding [9,10,11,12,30]. In our study, the overall rebleeding rate, when defined as any bleeding following initial haemostasis, was 37%, however the 30-day rebleeding rate, which is likely more representative of the efficacy of treatment delivered, was 29%. Possible explanations for this slightly higher rebleeding rate include a notably longer follow-up period (median 48 months) in comparison to other published studies which is likely to include significantly delayed rebleeding not captured by these. Our cohort included a high number of patients with severe liver disease (Child-Pugh score B or C) which has been shown to be risk factor for rebleeding [35], and while not statistically significant, there was a tendency towards higher rebleeding rates in these groups. In those patients with gastric varices who did re-bleed, however, 56% were successfully managed with further endoscopic therapy with thrombin. This demonstrates that thrombin is an effective therapy both as a bridging agent if interventional radiology is unavailable or unsuitable, and also as a definitive therapy when used alone.

No significant adverse events or complications were reported secondary to thrombin injection in the duration of our study. This emphasises how safe thrombin is as a treatment modality, avoiding the documented risks of local and embolic complications seen with glue.

This study also demonstrates the use of thrombin as primary therapy or adjunctive treatment for bleeding oesophageal varices (OV), ulcers secondary to EBL and ectopic varices. Whilst there are well established guidelines for the management of OV bleeding, EBL-induced ulceration and ectopic varices often present a management challenge. The prevalence of EBL-induced ulcer bleeding is reported to be 3.6–15% [36,37,38] with high mortality rates due to rebleeding, and management is highly variable [39,40]. Ectopic varices can involve any site within the GI tract, most commonly the rectum or duodenum. Whilst they are rare, haemorrhage can be significant if it occurs and the optimal management strategy again is unclear [41,42]. Our study demonstrates efficacy of thrombin in the management of bleeding from these indications. There was no significant difference in the initial haemostasis, rebleeding rates or mortality when comparing the gastric varices group to the groups receiving thrombin for EBL-induced ulceration or ectopic variceal bleeding. However, the analysis was limited by small sample sizes in these groups.

The clinical role of thrombin in our centre is as the primary endoscopic therapy for bleeding gastric varices in preference to glue due to familiarity with this technique and its excellent safety profile. As mentioned above it is also used as an adjunctive therapy for difficult to manage OV, EBL-induced ulcers and ectopic variceal bleeding, and is a very useful technique to have available for these scenarios. The expertise required is mainly regarding the preparation of the thrombin, as the injection technique is very simple, especially compared with the complexity and risk of glue injection.

The main limitation of this study is that it was a retrospective observational study. Therefore, our results cannot be compared with prospective clinical trial data and we cannot directly compare the efficacy of thrombin with glue or other treatment modalities. In its favour it could be argued that this is a real-world study and patients had a lengthy follow-up compared with previous studies. The long follow-up data adds to the validity of our observations especially as thrombin is a less well-studied treatment option and may account for the relatively high rebleeding rate in our group. Prospective, randomised trials comparing thrombin to glue and to IR techniques such as TIPSS and BRTO are required to contribute high quality evidence to this area.

## 5. Conclusions

In conclusion, this study represents to our knowledge one of the largest cohorts to date reporting the use of thrombin in the management of gastric, oesophageal and ectopic variceal haemorrhage. Thrombin is shown to be an effective treatment option for the above indications. Considering that thrombin treatment is easy to apply and safe, it should be considered as first line endoscopic therapy for control of bleeding gastric and ectopic varices. Further, larger randomised controlled trials are required to directly compare thrombin with other therapeutic options.

## Figures and Tables

**Figure 1 jcm-10-00785-f001:**
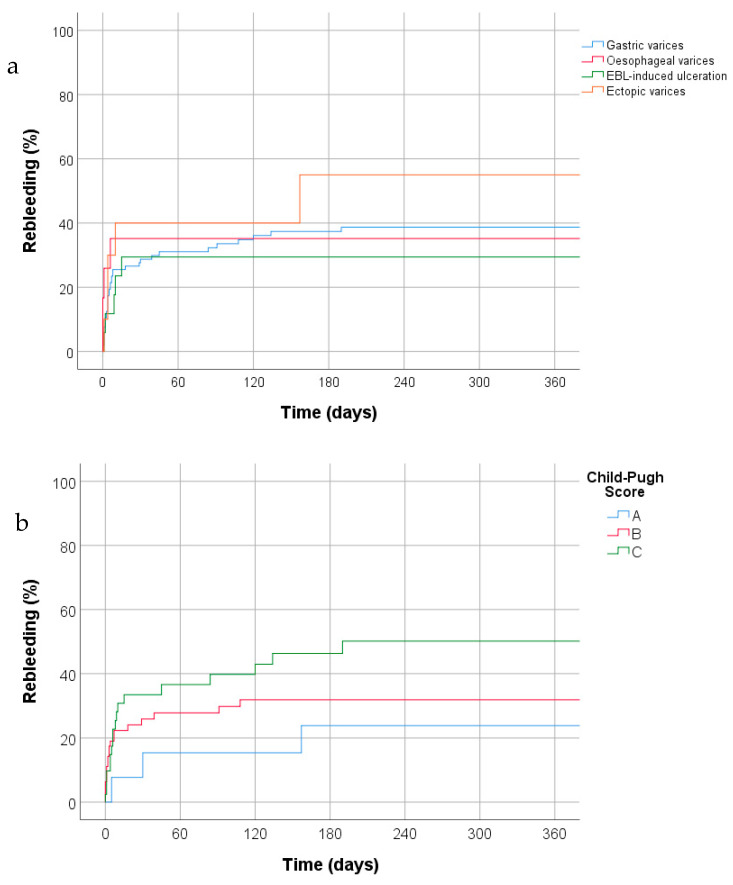
Cumulative rebleeding rates (Kaplan Meier plots) by (**a**) indication for thrombin injection and (**b**) Child-Pugh Score.

**Figure 2 jcm-10-00785-f002:**
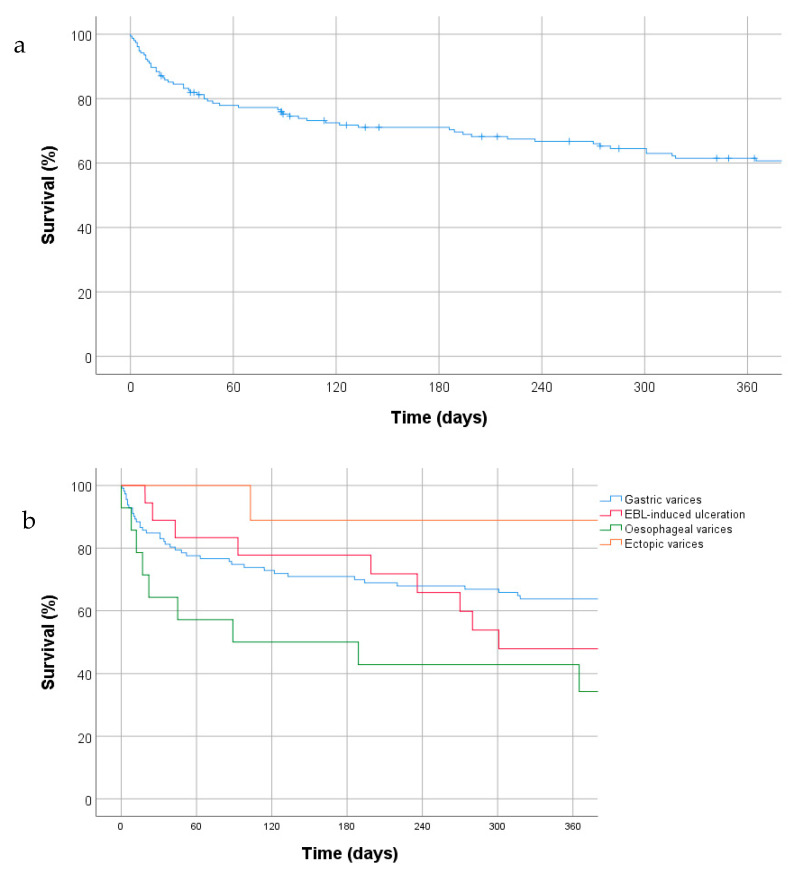
Cumulative survival rates (Kaplan–Meier plots) in our patient group (**a**) for the overall cohort and (**b**) by indications for thrombin injection.

**Table 1 jcm-10-00785-t001:** Patient characteristics (*n* = 155).

Demographic Details				
Gender	Male			96 (62%)
	Female			59 (38%)
Age (years)	Median (range ± SD)			58 (27–84, 12.8)
Aetiology of portal hypertension				
Cirrhosis	138 (89%)	MELD mean/median (range ± SD)		18.7/16 (7–40,7.7)
		Child-Pugh Score	A	15 (11%)
			B	65 (47%)
			C	50 (36%)
			Unavailable	8 (6%)
		Aetiology	ARLD	96 (70%)
			NAFLD	19 (14%)
			HCV	7 (5%)
			PBC	4 (3%)
			PSC	3 (2%)
			AIH	3 (2%)
			Cryptogenic	3 (2%)
Non-cirrhotic portal HTN	16 (10%)		PVT or SVT	15 (94%)
			NRH	1 (6%)
No evidence of portal HTN	1 (1%)		Gastric cancer	1

ARLD–alcohol-related liver disease; NAFLD–non-alcoholic fatty liver disease; HCV–hepatitis C infection; PBC–primary biliary cholangitis; PSC–primary sclerosing cholangitis; AIH–Autoimmune hepatitis; PVT–portal vein thrombosis; SVT–splenic vein thrombosis; NRH–nodular regenerative hyperplasia. SD—standard deviation.

**Table 2 jcm-10-00785-t002:** Indications for thrombin injection.

Indication for Thrombin Injection Therapy	No of Patients
Gastric varices	112 (72.2%)
GOV1	40
GOV2	51
IGV1	11
IGV2	10
EBL-induced ulceration	19 (12%)
Oesophageal Varices	14 (9%)
Rectal Varices	5 (3.2%)
Stomal Varices	3 (1.9%)
Duodenal Varices	2 (1.2%)

Gastric varices classified as per Sarin classification. EBL–endoscopic band ligation.

**Table 3 jcm-10-00785-t003:** Outcomes by presence of cirrhosis.

Cirrhotic	Yes (*n* = 138)	No (*n* = 17)	*p*-Value
Initial haemostasis	127 (92.0%)	17 (100%)	(*p* value unavailable as chi-square test could not be carried out for this dataset)
Rebled	47 (34.1%)	8 (47.1%)	0.29
Time to rebleed			
Early (<5 days)	22 (46.8%)(2 patients in this group did not have initial haemostasis)	6 (75.0%)	*n*/a (as above)
Late (5–29 days)	13 (27.7%)	2 (25.0%)	
Delayed (≥30 days)	12 (25.6%)	0	
Cause of index bleed			
GOV	96 (69.6%)	16 (94.1%)	*n*/a (as above)
OV	14 (10.1%)	0	
Banding ulcers	19 (13.8%)	0	
Ectopic varices	9 (6.5%)	1 (5.9%)	

GOV–gastro-oesophageal varices; OV-oesophageal varices.

**Table 4 jcm-10-00785-t004:** Outcomes by Child-Pugh Score.

Child-Pugh Score	A (*n* = 15)	B (*n* = 65)	C (*n* = 50)	*p*-Value
Cause of index bleed				
Gastric ± oesophageal varices	12 (80%)	50 (76.9%)	29 (58%)	0.42
Isolated oesophageal varices	0	6 (9.2%)	8 (16%)	
EBL-related ulceration	2 (13.3%)	6 (9.2%)	11 (22%)	
Ectopic varices	1 (6.7%)	3 (4.6%)	2 (4%)	
Re-bleeding data				
No. of pts where initial haemostasis with thrombin was achieved	14 (93.3%)	63 (96.9%)	42 (84%)	0.046significance is between B and C
Re-bleeding in pts who had initial haemostasis	3/14 (21.4%)	20/63 (31.7%)	19/42 (45.2%)	0.19
Early rebleeding (<5 days)	0	13	7	0.48 (rebleeding <30 days vs. delayed rebleeding vs. no rebleeding) 0.28 comparing rebleeding alone
Late rebleeding (5–29 days)	1	4	7	
Delayed rebleeding (≥30 days)	2	5	5	
Causes of re-bleeding				
Ulcers (including banding-related)	1 (33.3%)	3 (15%)	7 (36.8%)	0.30
Varices	1 (33.3%)	14 (70%)	7 (36.8%)	
Other	1 (33.3%)	-	-	
Unclear/no data/ not re-scoped	-	3 (15%)	5 (26.3%)	
Outcomes (alive/dead at time of follow-up)				
Alive	10 (66.7%)	28 (43.1%)	14 (28%)	0.021
Dead	5 (33.3%)None in rebleeding grp	37 (56.9%)14 in rebleeding grp	36 (72%)17 in rebleeding grp	
Death directly caused by liver disease	3/5 (60%)	26/37 (70.3%)	29/36 (80.6%)	0.45

## Data Availability

The data presented in this study are available on request from the corresponding author.

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
