# Peer review of "Thrombin Is an Effective and Safe Therapy in the Management of Bleeding Gastric Varices. A Real-World Experience"

_jcm, 2021, doi:10.3390/jcm10040785_

Round 1

Reviewer 1 Report

This retrospective observational study addresses the efficacy and safety of human recombinant thrombin in the management of hemorrhage caused by gastric/oesophageal/ectopic varices and endoscopic band ligation-induced ulceration.  The manuscript is concise and well-written. The main limit is that there is no comparison with other treatments. I have no suggestions for revision.

Author Response

No revisions required.

Reviewer 2 Report

  1. This type of research was done in the past, can you comment why is this relevant in the era when TIPS is widely available and is the standard of care.
  2. Can you comment on why only 28% had a TIPS
  3. Comment on the side effects of injecting rectal varices
  4. In the survival curve, can you include the patients who had a TIPS
  5. Describe the method how to "mix" the thrombin

Author Response

Response to reviewer 2

  1. This type of research was done in the past, can you comment why is this relevant in the era when TIPS is widely available and is the standard of care.

AND

  1. Can you comment on why only 28% had a TIPS

Response:

Our cohort of patients were an unfiltered group all presenting with acute GI bleeding and therefore in the first instance required emergency upper GI endoscopy (or flexible sigmoidoscopy in a very small number of cases) to determine the source of bleeding.  In a proportion it was the first presentation of cirrhosis/portal hypertension.

In 63.2% of patients, initial endoscopic therapy with thrombin (+/- other modalities such as EBL in some cases) was successful with no rebleeding.  Therefore in this group there would be no definite indication for TIPSS.

Whilst 24-hour interventional radiology is available at our centre, many centres do not have access to this, and as emergency endoscopy is essential first line to assess the source of bleeding and ideally achieve haemostasis, the study of potential endoscopic therapeutic options is still vitally important.

As you mention, only 28% of patients with gastric varices who rebled had a TIPSS procedure in our study.  3 patients underwent splenic artery embolization and 22 (56%) received successful endoscopic treatment with thrombin.  Some patients in this group were unsuitable for TIPSS either due to splenic or portal vein thromboses or the presence of hepatic encephalopathy.

Revisions to paper:

The other main approach to acutely bleeding gastric varices is interventional radiology (IR), primarily transjugular portosystemic stent-shunt (TIPSS); this is generally reserved for bleeding refractory to endoscopic therapy or for secondary prevention of rebleeding. (8)  As not all centres have 24-hour access to interventional radiology, and these techniques are not technically possible in all patients, endoscopic therapeutic options remain vitally important. (Lines 51 – 56)

Three patients (7.7%) underwent splenic artery embolization.  Reasons for attempting further endoscopic therapy rather than TIPSS in patients who rebled included the presence of splenic or portal vein thromboses making the procedure technically challenging, and the presence of hepatic encephalopathy. (Lines 209 – 212)

  1. Comment on the side effects of injecting rectal varices

Response:

No adverse effects occurred in the 5 patients treated with thrombin for rectal varices in our study.  There is very limited published data on the use of thrombin for rectal varices however no adverse effects have been noted. (1, 2)

Given the small number in this group and lack of specific adverse events we have included this in a statement on adverse effects for all groups.

Revisions to paper:

There were no reported clinically significant adverse events or complications resulting from thrombin therapy for any indication over the duration of follow up in our patient group. (Lines 199)

References:

  1. Robertson, M., Thompson, A.I. & Hayes, P.C. The Management of Bleeding from Anorectal Varices. Curr Hepatology Rep16, 406–415 (2017). https://doi.org/10.1007/s11901-017-0382-6
  2. McAvoy NC, Plevris JN, Hayes PC. Human thrombin for the treatment of gastric and ectopic varices. World J Gastroenterol. 2012;18(41):5912-5917. doi:10.3748/wjg.v18.i41.5912

  1. In the survival curve, can you include the patients who had a TIPS

Response:

We have limited the study to focus on the outcomes from the use of thrombin therapy and therefore these patients are included as part of the whole cohort outcomes and by indication for therapy.

The patients who had a TIPSS in this study primarily underwent this as a rescue procedure for rebleeding, although a small number had one placed electively for secondary prevention and are therefore a self-selecting high-risk group.

It is a heterogenous group with placement of TIPSS at different timepoints in their presentations and I think it may be very difficult/impossible to therefore compare this to give a meaningful result.

  1. Describe the method how to "mix" the thrombin

Each 2500IU vial of thrombin powder was reconstituted with 10mL sodium chloride 0.9% by gently swirling until completely dissolved, to give a final concentration of 250IU/mL (Lines 109-111)

Reviewer 3 Report

The total number of variceal bleeding cases including by other methods, such as TIPSS or BRTO or glue injection during the study-period should be described. Detailed methods on endoscopic hemostasis, such as injection speed of thrombin, diameter of injection needle, and procedure in time of  bleeding from puncture site.

Author Response

     1. The total number of variceal bleeding cases including by other methods, such as TIPSS or BRTO or glue injection during the study-period should be described.

Response:

The study population was derived from a database of all patients receiving thrombin for variceal bleeding during the study period as the focus of the study was to describe the outcomes for patients treated with this modality.  Full data for patients presenting with variceal bleeding treated by other modalities are unfortunately not available. 

  1. Detailed methods on endoscopic hemostasis, such as injection speed of thrombin, diameter of injection needle, and procedure in time of bleeding from puncture site.

Each 2500IU vial of thrombin powder was reconstituted with 10mL sodium chloride 0.9% by gently swirling until completely dissolved, to give a final concentration of 250IU/mL. Thrombin was injected using a standard 23GA sclerotherapy needle directly into the varix or bleeding banding ulcer.  The needle was held in place for up to 10 seconds before retracting to reduce the risk of bleeding from the puncture site. (Lines 109-114)

Reviewer 4 Report

dear authors

the topic of your paper is really interesting and stimulating

1)The main drawbacks are: it is a retrospective one, so underline strongly this point

2) in the literature review that I conducted not so many papers are present on this topic: so discuss what are the possible reasons of this

and discuss also about the cost of thrombin: may be a possible reason of scarse diffusion of this therapy

Author Response

1)The main drawbacks are: it is a retrospective one, so underline strongly this point

Revisions to paper:

The main study limitations are its retrospective nature and that it is non-comparative with other available treatment modalities, so direct comparisons of the efficacy of thrombin cannot be made. (Lines 354-356)

Prospective, randomised trials comparing thrombin to glue and to IR techniques such as TIPSS and BRTO are required to contribute high quality evidence to this area. (Lines 360-362)

2) in the literature review that I conducted not so many papers are present on this topic: so discuss what are the possible reasons of this

and discuss also about the cost of thrombin: may be a possible reason of scarse diffusion of this therapy

Yes you are very correct that there is limited published data on the use of thrombin, particularly in comparison with other modalities such as glue.  This is the reason why we feel our study is of importance as it is the largest cohort published to date describing the use of thrombin for gastric and ectopic varices.  Many centres are more comfortable with the use of glue, which is currently first line therapy as advised by the BSG guidelines and have less experience of thrombin.  In addition, in our centre, thrombin is used off-license from part of a haemostatic matrix kit (Floseal) so this may contribute to it being more difficult to obtain at an individual health board level.

Revisions to paper:

Large studies regarding the effectiveness and safety of thrombin are lacking, and injection therapy with glue is favoured by many centres, resulting in less familiarity with thrombin. (Line 48)

The per unit price for this kit (as of 2021) was £287.83. (Line 109)